# The Role of Rosavin in the Pathophysiology of Bone Metabolism

**DOI:** 10.3390/ijms25042117

**Published:** 2024-02-09

**Authors:** Piotr Wojdasiewicz, Paweł Turczyn, Anna Lach-Gruba, Łukasz A. Poniatowski, Daryush Purrahman, Mohammad-Reza Mahmoudian-Sani, Dariusz Szukiewicz

**Affiliations:** 1Department of Biophysics, Physiology and Pathophysiology, Faculty of Health Sciences, Medical University of Warsaw, Chałubińskiego 5, 02-004 Warsaw, Poland; piotr.wojdasiewicz@wum.edu.pl; 2Department of Early Arthritis, Eleonora Reicher National Institute of Geriatrics, Rheumatology and Rehabilitation, Spartańska 1, 02-637 Warsaw, Poland; pawel.turczyn@spartanska.pl; 3Department of Rehabilitation, St. Anna’s Trauma Surgery Hospital, Mazovian Rehabilitation Center—STOCER, Barska 16/20, 02-315 Warsaw, Poland; lach.gruba@onet.pl; 4Department of Neurosurgery, Dietrich-Bonhoeffer-Klinikum, Salvador-Allende-Straße 30, 17036 Neubrandenburg, Germany; lukasz.poniatowski@gmail.com; 5Thalassemia and Hemoglobinopathy Research Center, Health Research Institute, Ahvaz Jundishapur University of Medical Sciences, Ahvaz, Iran; daryushpurrahman@gmail.com (D.P.); mohamadsani495@gmail.com (M.-R.M.-S.)

**Keywords:** rosavin, adaptogens, bone metabolism, osteoblasts, osteoporosis, *Rhodiola rosea*, bone loss, osteoclastogenesis, osteogenesis

## Abstract

Rosavin, a phenylpropanoid in *Rhodiola rosea*’s rhizome, and an adaptogen, is known for enhancing the body’s response to environmental stress. It significantly affects cellular metabolism in health and many diseases, particularly influencing bone tissue metabolism. In vitro, rosavin inhibits osteoclastogenesis, disrupts F-actin ring formation, and reduces the expression of osteoclastogenesis-related genes such as cathepsin K, calcitonin receptor (CTR), tumor necrosis factor receptor-associated factor 6 (TRAF6), tartrate-resistant acid phosphatase (TRAP), and matrix metallopeptidase 9 (MMP-9). It also impedes the nuclear factor of activated T-cell cytoplasmic 1 (NFATc1), c-Fos, the nuclear factor kappa-light-chain-enhancer of activated B cells (NF-κB), and mitogen-activated protein kinase (MAPK) signaling pathways and blocks phosphorylation processes crucial for bone resorption. Moreover, rosavin promotes osteogenesis and osteoblast differentiation and increases mouse runt-related transcription factor 2 (Runx2) and osteocalcin (OCN) expression. In vivo studies show its effectiveness in enhancing bone mineral density (BMD) in postmenopausal osteoporosis (PMOP) mice, restraining osteoclast maturation, and increasing the active osteoblast percentage in bone tissue. It modulates mRNA expressions by increasing eukaryotic translation elongation factor 2 (EEF2) and decreasing histone deacetylase 1 (HDAC1), thereby activating osteoprotective epigenetic mechanisms, and alters many serum markers, including decreasing cross-linked C-telopeptide of type I collagen (CTX-1), tartrate-resistant acid phosphatase 5b (TRACP5b), receptor activator for nuclear factor κ B ligand (RANKL), macrophage-colony-stimulating factor (M-CSF), and TRAP, while increasing alkaline phosphatase (ALP) and OCN. Additionally, when combined with zinc and probiotics, it reduces pro-osteoporotic matrix metallopeptidase 3 (MMP-3), interleukin 6 (IL-6), and tumor necrosis factor α (TNF-α), and enhances anti-osteoporotic interleukin 10 (IL-10) and tissue inhibitor of metalloproteinase 3 (TIMP3) expressions. This paper aims to systematically review rosavin’s impact on bone tissue metabolism, exploring its potential in osteoporosis prevention and treatment, and suggesting future research directions.

## 1. Introduction

*Rhodiola rosea* belongs to the Crassulaceae plant family. It has been used in traditional medicine for over 2000 years as an adaptogen, increasing the body’s immunity and adaptive capabilities in unfavorable environmental conditions. *Rhodiola rosea* stimulates the nervous system, improves physical endurance, reduces fatigue, and has antidepressant and anti-inflammatory properties [1,2,3,4,5]. This plant requires specific growing conditions and occurs naturally in specified regions of the Earth, especially in Northern Europe, Northern Asia (Siberia), and North America [6]. Large commercial cultivations of this plant are currently present in China and Ukraine. Due to many years of observation of the health benefits of *Rhodiola rosea*, especially its root extracts, it has become the subject of numerous scientific studies in the fields of biochemistry, physiology, pathophysiology, and medicine. Phytochemical studies have identified approximately 140 chemical compounds in *Rhodiola rosea* roots, divided into six groups: phenylpropanoids, phenylethanoid derivatives, flavonoids, monoterpene derivatives, triterpenes, and phenolic acids [7]. Subsequent studies have shown that the health-promoting properties of *Rhodiola rosea* roots are primarily attributed to rosavin, a phenylpropanoid, and salidroside, a phenylethanoid derivative [8]. Initially, assessing the individual impact of each compound on human health was difficult. Recent research trends include laboratory studies in cell cultures and animal models assessing the independent effects of salidroside and rosavin on various physiological and pathophysiological processes, including cellular metabolism, inflammation, and regeneration. The main scientific community interest is more oriented toward salidroside, considered to be the main and most characteristic representative of the group of phenylethanoid derivatives, next to the well-studied tyrosol [9,10]. Nevertheless, in the latest studies, rosavin, one of several phenylpropanoids found in *Rhodiola rosea* roots, along with others, such as rosin and rosarin, is also gaining attention in academic investigations. Studies have confirmed the neuroprotective effect of rosavin in animal models of Alzheimer’s disease [11] and in cerebral ischemia [12], its role in limiting respiratory tract damage in sepsis [13], its anticancer effect in small-cell lung cancer [14], and its anti-inflammatory effect in fibrosis liver [15] and acute colitis [16]. Additionally, it has been demonstrated that rosavin affects bone tissue metabolism by various means, including the limitation of osteoclastogenesis and the augmentation of osteoblast numbers. Rosavin also engages in immunologic processes at the cellular level, impacting both cellular and epigenetic levels. The mentioned processes are anticipated to exert both indirect and direct impacts on enhancing the mechanical resilience of bone by strengthening its architecture, including the maintenance of a proper trabecular lacunae morphology and the woven structure of cancellous bone [17,18]. With the continually rising number of patients affected by degenerative bone diseases, such as osteoporosis, osteoarthritis, and rheumatoid arthritis, it is crucial to explore new osteoprotective molecules that could potentially become the standard of care for individuals with reduced bone mineral density (BMD) [19,20,21,22]. For instance, patients with osteoporosis in Europe, Canada, and the USA face annual healthcare costs ranging from USD 5000 to 6500 billion, specifically attributable to osteoporotic fractures [23,24]. The comprehensive figures encompassing losses for the healthcare system and the economy due to osteoporotic disability or impaired productivity in professional life are staggering [23]. Based on the gathered literature, the authors of this paper argue that rosavin may serve as a potential molecule with a beneficial effect on bone tissue metabolism, thereby contributing to an improved prognosis in individuals with osteoporosis and other metabolic bone diseases, while also potentially reducing associated treatment costs. It is worth noting that rosavin, as a phytochemical compound, is relatively cost-effective compared with laboratory drugs and can be administered orally, providing an additional advantage in terms of possible widespread ease of intake. The aim of this work is to critically systematize the current knowledge about the influence of rosavin on the metabolism of bone tissue, considering potential future directions of research in this area.

## 2. Biochemical Structure of Rosavin

Rosavin is a phytochemical belonging to the group of phenylpropanoids [1]. This group of chemical compounds contains a six-carbon aromatic phenyl group and a three-carbon propene tail. Phenylpropanoids are synthesized from the amino acid phenylalanine via the shikimate pathway, which primarily occurs in bacteria, fungi, and plants (the phenylpropanoid synthetic pathway does not occur in animals) [25]. The chemical formula of rosavin is C_20_H_28_O_10_ [26], and its molecular weight is 428.4 g/mol [27]. In its solid form, rosavin appears as a white powder and is characterized by good solubility in water. The chemical structure of rosavin is depicted in Figure 1.

## 3. Impact of Rosavin on Bone Tissue Metabolism

In the study conducted by Zhang et al., the aim was to investigate the effect of rosavin on osteoclastogenesis in mouse models of ovariectomy-induced osteoporosis (OVX), which is analogous to postmenopausal osteoporosis (PMOP), as well as in in vitro cell culture conditions [28]. For in vitro tests, standard cell models for osteoclastogenesis were used, including bone marrow mesenchymal stem cells (BMMSCs) and murine macrophage cells (RAW264.7 cells). Five cell colonies were created for each cell type: the first colony served as the control culture, receiving only a dedicated medium. The remaining four colonies were exposed to receptor activator for nuclear factor κ B ligand (RANKL) (100 ng/mL) and macrophage-colony-stimulating factor (M-CSF) (30 ng/mL). Additionally, three of these colonies received supplementation with increasing concentrations of rosavin (1.25 μM, 2.5 μM, and 5 μM respectively). After seven days of incubation, the morphology of the cell colonies and the number of tartrate-resistant acid phosphatase-positive cells (TRAP-positive cells), which in this study corresponded to osteoclasts, were evaluated. It was observed that with increasing concentrations of rosavin, the number of TRAP-positive cells decreased, and at a concentration of 5 μM, the morphology of both BMMSCs and RAW264.7 colonies closely resembled that of the control colony [28]. Subsequent analyses confirmed the positive impact of rosavin on osteogenesis in BMMSC cultures. After 20 days of incubation with rosavin, alkaline phosphatase (ALP) staining and alizarin red staining were performed, which were used to identify osteoblasts in the examined tissue. Compared with the control group, it was demonstrated that rosavin significantly stimulated osteogenesis by increasing the population of osteoblast cells [28]. BMMSC cultures with rosavin also exhibited increased expression of mouse runt-related transcription factor 2 (Runx2) and osteocalcin (OCN), which are widely recognized as typical markers of ongoing osteogenesis [29,30]. In the same study, it was noted that as the concentration of rosavin increased in the respective BMMSC cultures, the number and size of the F-actin rings decreased. The formation of F-actin rings is observed during osteoclastogenesis and is a physiological process in this cell line, without which the bone resorption process cannot proceed [31]. The reduced formation of F-actin rings, resulting from the action of rosavin, affects bone tissue remodeling by hindering the catabolic function of osteoclasts. Simultaneously, it promotes cellular divisions and the differentiation of the osteoblastic lineage [31]. This is schematically presented in Figure 2. The influence of rosavin, directly proportional to its concentration, was also evident in the reduced surface area of resorbed bone tissue compared with the control group that did not receive rosavin supplementation [28]. Zhang et al. also demonstrated that rosavin inhibits osteoclastogenesis most effectively in BMMSCs and RAW264.7 colonies when administered to cells one day after the addition of RANKL and M-CSF [28]. The effectiveness of osteoclastogenesis inhibition decreased with a delayed intervention of rosavin in these colonies. Rosavin added on the third day in the case of the RAW264.7 colonies, and on the fifth day in the case of BMMSCs, no longer inhibited osteoclast formation. Western blot analysis confirmed these observations [28]. A greater decrease in the expression levels of osteoclastogenesis-related genes, such as cathepsin K, calcitonin receptor (CTR), tumor necrosis factor receptor-associated factor 6 (TRAF6), TRAP, and matrix metallopeptidase 9 (MMP-9), was noted in colonies that received rosavin on the first day after the administration of RANKL and M-CSF compared with those receiving it on the third and fifth days [28]. From other studies on bone tissue metabolism, it is known that the reduction in the expression of these genes in bone tissue is due to a decrease in the cellular expression of nuclear factor of activated T-cell cytoplasmic 1 (NFATc1), which is considered a key transcription factor determining osteoclastogenesis [32,33]. Blocking NFATc1 inhibits osteoclast maturation, rendering precursor cells insensitive to the action of RANKL [34,35]. In this study, rosavin reduced the presence of NFATc1 in BMMSCs in a manner directly proportional to the concentration, following the principle that “the earlier the intervention, the more effective”, which correlates with the aforementioned results of rosavin’s osteoprotective effect on bone tissue. Surprisingly, rosavin did not show any changes in the expression levels of receptor activator of nuclear factor-κB (RANK) and colony-stimulating factor-1 receptor (c-fms) in RAW264.7 colonies treated with M-CSF (which was determined using real-time quantitative reverse transcription–polymerase chain reaction (qRT-PCR) [28]. Increased expression of these receptors is observed in osteoporosis and is responsible for intensifying pathological processes related to osteoclastogenesis [36,37]. Therefore, in this particular case, rosavin did not exhibit a potential inhibitory effect on osteoclastogenesis. Another analysis revealed that rosavin inhibits the signaling pathways induced by RANKL, which are typical in osteoclastogenesis and osteoporosis, including the nuclear factor kappa-light-chain-enhancer of activated B cells (NF-κB) [38,39] and mitogen-activated protein kinase (MAPK) pathways [40]. An amount of 5 μM of rosavin leads to the inhibition of the cellular phosphorylation of human protein 56 (p56) and p56 translocation into the nucleus of RAW264.7 cells, blocking the biological effect of the NF-κB signaling pathway and osteoclastogenesis [28]. Rosavin also blocks the MAPK pathway by inhibiting the phosphorylation of its components, including extracellular-signal-regulated kinase (ERK), human protein 38 (p38), and c-Jun N-terminal kinase (JNK), which may have potential significance in the treatment of osteoporosis [28]. Next to these findings, the inhibition of the MAPK pathway by rosavin was also indirectly observed. An effect of the MAPK pathway was the increased cellular expression of the proto-oncogene c-Fos. Rosavin significantly reduced the expression of c-Fos in RAW264.7 cells stimulated with RANKL, providing indirect evidence of the effectiveness of blocking the MAPK pathway, which is responsible for bone tissue resorption [28]. A somewhat surprising result of the above-mentioned analysis is rosavin’s inhibition of the phosphorylation of the inhibitory subunit of NF-κB alpha (IκBα). Phosphorylated IκBα is responsible for inhibiting the NF-κB signaling pathway. Excess amounts of the dephosphorylated form of IκBα result in the activation of the NF-κB signaling pathway. However, as it turns out, this action has weaker biological effects than inhibiting p56 phosphorylation, so it does not change the final qualification of rosavin’s metabolism manifestation, which is the inhibition of bone tissue resorption [28]. Mice with OVX-induced osteoporosis, which is analogous to PMOP, are a standard animal model in osteoporosis studies [41]. On such an animal model, another study was conducted to assess the effect of rosavin on bone tissue. The experimental group of mice received daily intraperitoneal injections of rosavin (10 mg/kg) for 6 weeks. Compared with the control group that did not receive the drug, rosavin led to an increase in BMD and trabecular bone surface in cross-sectional views of the distal femur and radiological studies. Bone loss in mice receiving rosavin was significantly limited [28]. This was confirmed by the TRAP staining of prepared histological specimens from the femurs, which showed a statistically significantly smaller number of TRAP-positive cells corresponding to osteoclasts. Rosavin also reduced the mice’s blood serum levels of markers of increased bone turnover, such as cross-linked C-telopeptide of type I collagen (CTX-1) and tartrate-resistant acid phosphatase 5b (TRAcp5b). In mice receiving rosavin, increased blood serum levels of ALP and OCN, which are considered serum bone formation markers by many researchers, were observed [42,43]. Rosavin’s osteoprotective action is also evident at the epigenetic level through the inhibition of the cellular activity of histone deacetylase 1 (HDAC1) [44]. HDAC1 belongs to the class of classical histone deacetylases that contain Zn^2+^ ions in the catalytic center. Based on many studies, it has been established that the overexpression of HDAC1 leads to the acetylation of many histones, resulting in the pathological silencing of the transcription of many genes and leading to the development of cancer or chronic inflammation [45,46,47]. Regarding bone tissue, it was observed that rosavin, by reducing the expression of HDAC1, increases the expression of eukaryotic translation elongation factor 2 (EEF2) in BMMSCs. Rosavin thus influences the inhibition of the deacetylation process of EEF2 [44]. EEF2, in turn, under physiological conditions, is responsible for bone formation processes and is postulated to have an anti-osteoporotic effect [46,48]. In a dedicated study on the HDAC1/EEF2 axis, it was observed that rosavin reduced the levels of RANKL, M-CSF, and TRAP in the blood serum of OVX mice [44]. In histological specimens of bone tissue from OVX mice receiving rosavin, there was a smaller number of osteoclasts, and the signs of bone resorption were less pronounced compared with OVX mice that did not receive intraperitoneal injections of rosavin. Furthermore, rosavin significantly increased the expression of bone formation markers such as ALP, OCN, and Runx2, as well as the number of osteoblasts (by increasing the expression of the surface marker for osteoblasts, osteoprotegerin (OPG)) in the bone tissue of OVX mice [44]. To confirm the involvement of rosavin in the HDAC1/EEF2 axis in osteoporosis, Zhang et al. decided to observe how the expressions of HDAC1 and EEF2 mRNA change in OVX mice and OVX mice that received rosavin [44]. It turned out that the mice that did not receive rosavin had increased mRNA expression for HDAC1 and decreased expression for EEF2, while the addition of rosavin reversed these effects [44]. The above observations became the basis for another in vitro study on the following cells: a mouse clonal, osteoblast-like cell line (MC3T3-E1) and BMMSCs (corresponding to osteoclasts). It was demonstrated that HDAC1 stimulates the proliferation of osteoclasts by reducing the duration of the G0/G1 phase (the acceleration of cell divisions) and inhibits cell divisions of osteoblasts. HDAC1 also stimulates the apoptosis of osteoblasts but inhibits the apoptosis of osteoclasts. Increased expression of EEF2 has the opposite effect of HDAC1 in BMMSCs and MC3T3-E1 cells, including inducing cell cycle arrest in the G0/G1 phase in osteoclasts, ultimately reducing their population [44]. A series of additional experiments on the MC3T3-E1 cell culture showed that rosavin increased the expression of EEF2 in these cells, resulting in an increased number of ALP-positive cells (corresponding to osteogenesis) and increased calcium nodules, corresponding to the activation of the calcium ion deposition process presented typically in bone formation [44]. In contrast, in the BMMSC culture, which received M-CSF and RANKL, rosavin reduced the expression of genes for HDAC1 and decreased the expression of typical promoters for osteoclastogenesis, such as TRAP, NFATc1, and cathepsin K [44]. This manifested itself in the inhibition of the differentiation and maturation processes of osteoclasts in the histological specimens observed. The impact on the HDAC1/EEF2 axis is a very promising potential direction for anti-osteoporotic drug action, also because this axis regulates intracellular signaling pathways for MAPK and NF-κB in the bone tissue of OVX mice [44]. The increased presence of EEF2 (in response to rosavin supplementation) in the bone tissue of OVX mice inhibits MAPK- and NF-κB-mediated osteoclastogenesis and reduces bone tissue resorption [44]. In summary, rosavin as a regulator of the HDAC1/EEF2 axis positively influences bone tissue metabolism, which may have potential significance in the treatment of reduced BMD. The scheme of rosavin’s action on the HDAC1/EEF2 axis with the biological effects on bone metabolism is presented in Figure 3. Oral supplementation with rosavin (100 mg/day) in combination with zinc (20 mg/day) and a probiotic complex (CNS, Pharm Korea Co., Ltd., Seoul, Republic of Korea) exhibited protective effects on rat bone tissue in a model of osteoarthritis (OA) induced by intra-articular (right knee) injection of 3 mg of monosodium iodoacetate compared with the control group and rats receiving only celecoxib (13.5 mg/day) [49]. Kwon et al. demonstrated this through the observation of post-mortem preparations of the distal femoral end of rats (by using micro-computed tomography (micro-CT)) receiving the aforementioned supplementation. The quantitative micro-CT results from the control group and celecoxib-treated group did not differ from each other and showed pronounced bone structure deterioration, confirmed by quantitative assessment parameters of bone tissue, such as object volume (Obj.V) and Obj.V/total volume (TV). Both of these parameters were higher and osteoporotic changes were inhibited in rats receiving rosavin with zinc and the probiotic complex. It is worth noting that besides osteoporosis, osteoarthritis often involves the thinning of the bone tissue structure, which is sometimes observed in radiological studies in the form of subchondral cysts [50,51]. Additionally, osteoarthritis during exacerbations is associated with increased expression of inflammatory cytokines in periarticular tissues (synovial membrane), such as interleukin 1β (IL-1β), tumor necrosis factor α (TNF-α), and interleukin 6 (IL-6), which, apart from inflammation, also stimulate osteoclastogenesis and bone resorption. This is significant due to the proximity between the synovial membrane and the subchondral bone tissue layer [52]. Although Kwon et al. did not analyze the isolated impact of rosavin on bone tissue quality, considering the knowledge gathered so far, the authors of this paper regard the results of the cited study as promising but certainly requiring further more detailed research. It is worth adding that Kwon et al.’s study also demonstrated the effectiveness of rosavin, zinc, and probiotic supplementation in reducing lower limb pain related to the musculoskeletal system and inhibiting inflammatory cytokines responsible for the development of both osteoarthritis and osteoporosis, i.e., matrix metallopeptidase 3 (MMP-3) IL-6, and TNF-α [49,52,53]. Furthermore, the aforementioned complex stimulates the secretion of anti-inflammatory interleukin 10 (IL-10) and tissue inhibitor of metalloproteinase 3 (TIMP3) by synovial membrane cells, which exhibit anabolic and protective effects on various tissues, including bone tissue [54,55].

## 4. Discussion

Despite the promising research results regarding the anti-osteoporotic, anabolic effects of rosavin on bone tissue, the number of publications addressing this issue remains limited. This is even more intriguing since osteoporosis is currently one of the major global health problems, incurring significant financial and social costs [19,20,21,23,24]. Therefore, it seems justified to make greater efforts to explore new methods of prevention or treatment for this disease. In the opinion of the authors of this paper, rosavin may prove to be a highly promising potential drug for osteoporosis. The literature gathered above strongly confirms rosavin’s involvement in the metabolic processes of bone tissue: inhibiting its catabolism and promoting anabolism. In in vitro studies, rosavin inhibits osteoclastogenesis, blocks the formation of F-actin rings, reduces the expression levels of osteoclastogenesis-related genes (cathepsin K, CTR, TRAF6, TRAP, and MMP-9), and inhibits the activity of NFATc1, c-Fos, and the NF-κB and MAPK intracellular signaling pathways [28]. Inhibition of the NF-κB signaling pathway occurs through rosavin’s inhibition of p56 phosphorylation and its translocation to the cell nucleus, while the reduced biological activity of the MAPK pathway results from the inhibition of ERK, p38, and JNK phosphorylation [28]. In terms of anabolism, rosavin stimulates osteogenesis by increasing the population of osteoblasts in cell cultures and increasing the expression of Runx2 and OCN [28]. The summary of rosavin’s biological manifestations in in vitro studies is presented in Table 1. In in vivo studies, rosavin increased the BMD of the distal end of the femur in a mouse model of PMOP, as confirmed radiologically and histologically. It reduced the number of osteoclasts while increasing osteoblasts in osteoporotic bone tissue. In the blood serum, rosavin decreased the levels of CTX-1, TRAcp5b, RANKL, M-CSF, and TRAP, and increased the levels of ALP and OCN [28]. In mouse bone tissue cells receiving rosavin, there was a decrease in HDAC1 mRNA expression and an increase in EEF2 mRNA expression (rosavin’s influence on epigenetic osteoprotective mechanisms) [44]. Additionally, an increase in the number of calcium nodules under microscope examination was observed [44]. In a rat model, rosavin in combination with zinc and probiotics reduced the expressions of MMP-3, IL-6, and TNF-α and increased the expressions of TIMP3 and IL-10 in the synovial membrane of knee joints. Rosavin also affected the inhibition of osteoporotic processes in bone tissue (these results were confirmed via quantitative micro-CT examination) [49]. The summary of rosavin’s biological manifestations in in vivo studies is presented in Table 2. It is worth noting that not all the research results cited in this study align with the osteoprotective profile of rosavin. This suggests that there is still much to discover about the multifaceted impact of rosavin on the complex network of metabolic and inflammatory processes within bone tissue. For example, rosavin inhibited the phosphorylation of the IκBα subunit in RAW264.7 cells, which is potentially responsible for activating the NF-κB signaling pathway, which is connected to osteoclastogenesis [28]. In the same cell culture, rosavin also did not decrease the expression of RANK and c-fms receptors, which is surprising taking into consideration its other anti-osteoporotic properties [28]. However, considering the cited publications, it should be acknowledged that rosavin’s osteoprotective action is probably significantly stronger overall than the individual examples promoting osteoclastogenesis. Moreover, it is not known what the resultant impact of these processes would be on other cell lines, and in a broader context, on the entire human system (i.e., potential side effects, systemic complications, etc.). In considering the potential use of rosavin as a treatment for osteoporosis, it is also important to determine whether it exhibits osteoprotective properties throughout the entire course of the disease. Some presented research results suggest that rosavin exhibits its reparative action on bone tissue metabolism and increases BMD more effectively the earlier it is administered (1 day > 3 days > 5 days) after the presence of inflammatory and osteoclastogenesis-promoting factors is discovered (i.e., RANKL and M-CSF) [28]. This may raise legitimate concerns about the effectiveness of rosavin in patients who have been suffering from osteoporosis for many years. In this situation, rosavin should be considered more as a potential preventive rather than a therapeutic measure. More detailed research is highly needed on this matter. It is worth noting that the biological manifestation of rosavin’s action in animal models occurred independently of its administration route; its influence on bone tissue metabolism was observed both through intraperitoneal injections and dietary supplementation [28,44,49]. The potential effectiveness of oral rosavin administration is another advantage, as it has been used in humans as an oral dietary supplement in the form of *Rhodiola rosea* extract successfully for hundreds of years. Oral administration is generally considered the most acceptable form of drug administration for patients compared with subcutaneous, intramuscular, or intravenous injections, according to many studies [56]. However, it should be noted that there is still insufficient knowledge in the international literature regarding rosavin’s ability to penetrate enterocytes through the endothelium into the bloodstream. We do not know exactly how rosavin’s chemical structure changes in the human gastrointestinal tract, what metabolism it undergoes after absorption into the bloodstream, and, more importantly, what therapeutic dose of rosavin administered orally or intravenously is optimal to show biological effects on bone tissue. So far, such studies have not been conducted, which are necessary in the context of any considerations regarding the use of rosavin as a useful compound in the treatment of osteopenia or osteoporosis [1]. Another aspect worth considering is the assessment of the collective interaction of rosavin and salidroside with bone tissue metabolism. Current international studies have focused on the individual impact of each of these substances on osteogenesis. However, it is worth noting that both of these compounds occur together in nature as components of *Rhodiola rosea* root and, as mentioned, have been taken together orally as a dietary supplement or adaptogens for many years [2,3,4]. Since research shows that rosavin and salidroside taken separately have anti-osteoporotic effects [57,58,59], it is necessary to determine whether the combined action of these substances undergoes an additive effect or whether some biological actions of rosavin and salidroside counteract each other. Conducting such studies would be valuable to gain a better understanding of the biological properties of rosavin and ways to potentially enhance its effectiveness by placing it in complex combinations with other substances, including salidroside. In the opinion of the authors of this paper, the next step toward establishing the potential osteoprotective action of rosavin on human bone tissue is to conduct research on human bone tissue cells, including osteoblasts and osteoclasts. So far, the cited results primarily concern animal models, which, while still promising, require further verification on human bone tissue. Such studies could determine safe rosavin doses in cell cultures and, more importantly, the therapeutic concentrations of rosavin that are optimal for human health. In light of all the gathered data, it is crucial to consider the potential therapeutic impact of rosavin on the prevention and treatment of complications associated with osteoporosis, particularly fractures. As BMD decreases and osteoclastogenesis increases in the course of osteoporosis, alterations occur in the microarchitecture of the bone, including the dysmorphic morphology of trabecular lacunae [60,61]. The emergence of such changes in the bone tissue structure could serve as a predictive factor for the rapid onset of osteoporotic fractures. An intriguing avenue for research would involve analyzing the influence of rosavin on changes in the microarchitecture of bone tissue, especially trabecular lacunae, in animal models of osteoporosis. The restoration of the lacunae and trabecular organization in bone tissue to a normal morphology could provide tangible evidence of the anabolic impact of rosavin on bone tissue metabolism. Additionally, with the advancement of high-resolution imaging techniques, it is anticipated that such observations could be made non-invasively in humans in the future. The authors of this paper believe that considering all the above information gathered in the literature review and discussion, rosavin should be regarded as a highly promising compound of plant origin with anti-osteoporotic properties. Rosavin has a multi-faceted protective effect on the metabolism of bone tissue, both at the level of cellular regulation of immunity, as well as through epigenetic processes. However, much more specialized laboratory and pharmacological tests are needed to consider this molecule useful in the prevention and treatment of osteoporosis in humans.

## Author Contributions

Conceptualization, P.W., P.T. and A.L.-G.; methodology, P.W., P.T., A.L.-G., Ł.A.P. and D.P.; software, P.W., P.T. and Ł.A.P.; validation, P.W., P.T., A.L.-G., Ł.A.P., D.P. and M.-R.M.-S.; formal analysis, P.W., A.L.-G., Ł.A.P., D.P. and M.-R.M.-S.; investigation, P.W., D.P., M.-R.M.-S. and D.S.; resources, Ł.A.P., D.P., M.-R.M.-S. and D.S.; data curation, P.W., P.T. and D.S.; writing—original draft preparation, P.W., P.T., A.L.-G., Ł.A.P. and D.P.; writing—review and editing, P.W., Ł.A.P., D.P., M.-R.M.-S. and D.S.; visualization, P.W., P.T. and A.L.-G.; supervision, P.W., P.T., Ł.A.P. and D.S.; project administration, P.W., Ł.A.P., M.-R.M.-S. and D.S.; funding acquisition, P.W., Ł.A.P., M.-R.M.-S. and D.S. All authors have read and agreed to the published version of the manuscript.

## Figures and Tables

**Figure 1 ijms-25-02117-f001:**
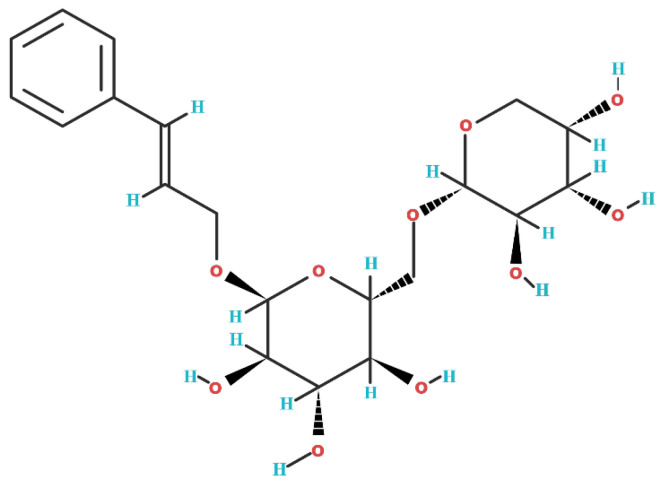
The two-dimensional chemical structure of rosavin (C_20_H_28_O_10_). Oxygen (O) atoms are denoted in red, and hydrogen (H) atoms are indicated in blue.

**Figure 2 ijms-25-02117-f002:**
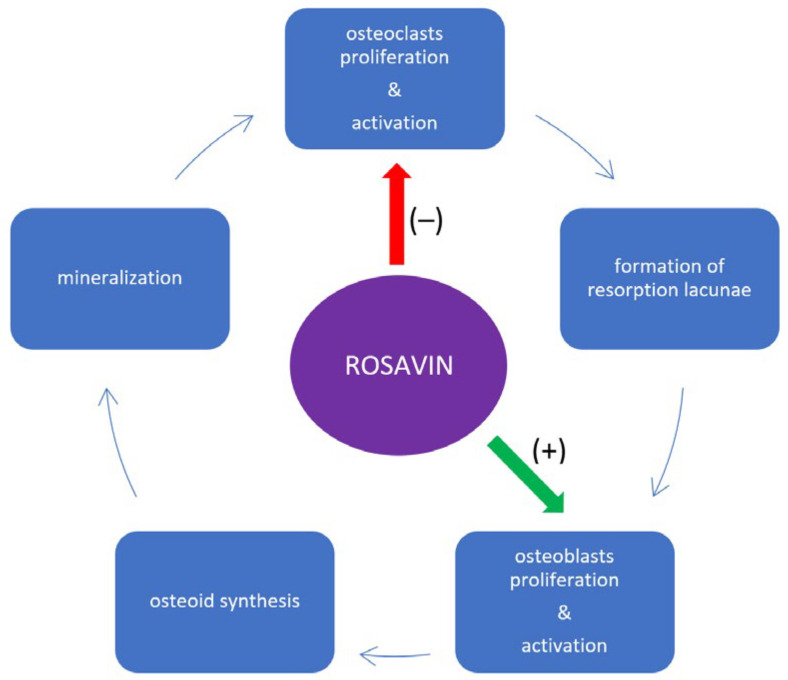
Schematic representation of the impact of rosavin on bone tissue remodeling. The bold red arrow with a (−) sign denotes an inhibitory effect, while the bold green arrow with a (+) sign signifies a stimulating effect.

**Figure 3 ijms-25-02117-f003:**
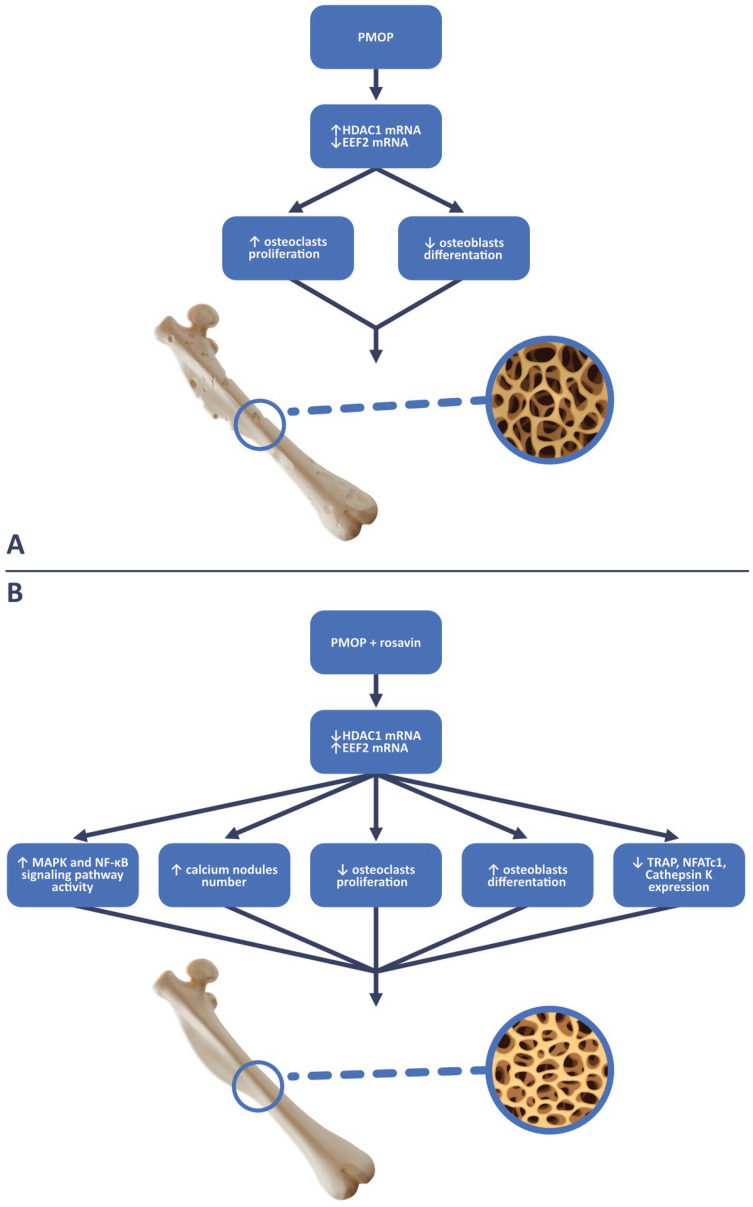
(**A**) A diagram illustrating epigenetic changes in the HDAC1/EEF2 signaling axis and their impact on bone tissue in the rodent postmenopausal osteoporosis (PMOP) model. (**B**) Influence of rosavin on the metabolism of murine bone tissue through the HDAC1/EEF2 signaling axis in ovariectomy (OVX) mice. The illustrations include three-dimensional models of the murine femoral bone, presenting both macroscopic and microscopic perspectives (trabecular bone tissue). ↑—increase, ↓—decrease, HDAC1—histone deacetylase 1, EEF2—eukaryotic translation elongation factor 2, MAPK—mitogen-activated protein kinase, NF-κB—nuclear factor kappa-light-chain-enhancer of activated B cells, TRAP—tartrate-resistant acid phosphatase, and NFATc1—nuclear factor of activated T-cells cytoplasmic 1.

**Table 1 ijms-25-02117-t001:** Biological manifestations of rosavin’s impact in in vitro studies on specific cell cultures [28,44]. (+)—stimulating/activating effect, (–)—inhibitory effect, (**↔**)—no effect, n/a—no data available, BMMSCs—bone marrow mesenchymal stem cells, RAW264.7—murine macrophage cells, MC3T3-E—a mouse clonal, osteoblast-like cell line, TRAP—tartrate-resistant acid phosphatase, OCN—osteocalcin, Runx2—mouse runt-related transcription factor 2, RANK—receptor activator of nuclear factor-κB, c-fms—colony-stimulating factor-1 receptor, CTR—calcitonin receptor, TRAF6—tumor necrosis factor receptor-associated factor 6, MMP-9—matrix metallopeptidase 9, NFATc1—nuclear factor of activated T-cells cytoplasmic 1, NF-κB—nuclear factor kappa-light-chain-enhancer of activated B cells, MAPK—mitogen-activated protein kinase, ERK—extracellular-signal-regulated kinase, p38—human protein 38, p56—human protein 56, JNK—c-Jun N-terminal kinase, IκBα—inhibitory subunit of NF-κB α, EEF2—eukaryotic translation elongation factor 2, and HDAC1—histone deacetylase 1.

	BMMSCs (Isolated from the Femoral and Tibial Bone Marrow of C57BL/6 Mice)	RAW264.7 (Murine Macrophage Cells)	MC3T3-E (ATCC, Manassas, VA, USA)
Osteoclastogenesis (number of TRAP-positive cells)	–	–	n/a
Osteoblast differentiation	+	n/a	+
Expressions of OCN and Runx2	+	n/a	n/a
Expressions of c-fms and RANK	**↔**	n/a	n/a
Formation of F-actin rings	–	n/a	n/a
Expressions of cathepsin K, CTR, TRAF6, TRAP, and MMP-9	–	–	n/a
Expression of NFATc1	–	–	n/a
Activity of NF-κB and MAPK signaling pathways	n/a	–	n/a
Expression of c-Fos	n/a	–	n/a
Phosphorylation of ERK, p38, p56, and JNK	n/a	–	n/a
Phosphorylation of IκBα	n/a	–	n/a
mRNA expression of EEF2	+	n/a	+
mRNA expression of HDAC1	–	n/a	–

**Table 2 ijms-25-02117-t002:** Biological manifestations of rosavin’s action in in vivo studies on blood serum, bone tissue, and synovial membrane depending on the animal species and form of rosavin administration [44,49]. ↑—increase, ↓—decrease, n/a—no data available, OVX—ovariectomy, CTX-1—cross-linked C-telopeptide of type I collagen, TRAcp5b—tartrate-resistant acid phosphatase 5b, RANKL—receptor activator for nuclear factor κ B ligand, M-CSF—macrophage colony-stimulating factor, TRAP—tartrate-resistant acid phosphatase, ALP—alkaline phosphatase, OCN—osteocalcin, EEF2—eukaryotic translation elongation factor 2, HDAC1—histone deacetylase 1, MMP-3—matrix metallopeptidase 3, IL-6—interleukin 6, TNF-α—tumor necrosis factor α, IL-10—interleukin 10, and TIMP3—tissue inhibitor of metalloproteinase 3.

	Blood Serum	Bone Tissue	Synovial Membrane
C57BL/6 and OVX mice (intraperitoneal injections of rosavin (10 mg/kg))	↓ concentrations of CTX-1, TRAcp5b, RANKL, M-CSF, and TRAP ↑ concentrations of ALP and OCN	↓ number of osteoclasts ↑ number of osteoblasts ↑ expression of EEF2 ↓ expression HDAC1 ↓ bone resorption ↑ number of calcium nodules	n/a
Male Wistar rats (Central Lab. Animal, Inc., Seoul, Korea) after intra-articular injection of monosodium iodoacetate (right knee) and oral administration of rosavin (100 mg/day) combined with zinc (20 mg/day) and a complex of probiotics (CNS, Pharm Korea Co., Ltd., Seoul, Korea)	n/a	↓ bone resorption	↓ expressions of MMP-3, IL-6, and TNF-α ↑ expressions IL-10 and TIMP3

## Data Availability

Data sharing is not applicable to this article.

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
