# Peer review of "The Role of Rosavin in the Pathophysiology of Bone Metabolism"

_ijms, 2024, doi:10.3390/ijms25042117_

Round 1

Reviewer 1 Report

Comments and Suggestions for Authors

The manuscript is an interesting analysis of the current state of knowledge regarding the effect of Rosavin on the metabolism of bone tissue, with particular emphasis on its osteoprotective effect. The authors describe the osteotropic effects of the use of Rosavin, including numerous examples of studies, both in vitro and in vivo. Although the possibilities of treating metabolic diseases of the skeletal system are very wide, the interest in preparations of natural origin does not wane. The manuscript under review follows this trend. This extremely interesting manuscript was prepared in a communicative way and linguistically correct. However, I have 3 comments, taking them into account, it may be published.

1. Page 3 instead of variectomy there should be ovariectomy

2. I believe that the current graphic form and font size in Fig. 2 make it illegible and require re-edition.

3. Tables 1 and 2 should be supplemented with reference numbers constituting the basis for their preparation. The table also includes different font sizes.

Reviewer 2 Report

Comments and Suggestions for Authors

The authors discuss the effects of rosavin, a phenylpropanoid found in the rhizome of Rhodiola rosea, on bone tissue metabolism and its potential role in preventing and treating osteoporosis.

While presenting potentially interesting applications, the paper should be further improved as follows:

1. Introduction: the focus of the paper is the study of the effects of rosavin on bone methabolism. However, the intricate complexity of bone architecture is not even mentioned (see https://doi.org/10.1007/s11517-021-02422-x and https://doi.org/10.1080/10255842.2021.1998465)

Besides, the potential effects of rosavin on bone lacunar level are not discussed. It is worth highlighting that several authors have recognized the role played by lacunar level mechanobiology for the early diagnosis of bone disorders (see https://doi.org/10.1016/j.matdes.2023.112087, and https://doi.org/10.1016/j.bone.2022.116424)

2. Impact of rosavin on bone tissue metabolism: in this paragraph, a figure related to bone remodelling processes and the role played by rosavin would be beneficial to ease the understanding.

3. Figure 2: please increase character size and increase the contrast. The figure is barely readable.

Reviewer 3 Report

Comments and Suggestions for Authors

Comments to the authors

Wojdasiewics et al are aiming to systematically review the potential role of rosavin, a unique phenylpropanoid in Rhodiola rosea’s rhizome, an adaptogen, on bone tissue metabolism. Recently, rosavin is gaining attention in scientific research. Many studies suggested the potential effect of rosavin for Alzheimer’s disease, lung cancer, and liver fibrosis. However, little is known about the potential effect of rosavin for bone diseases such as osteoporosis. Therefore, it is critical to review what we have known so far about the role of rosavin in bone field. Overall, this manuscript is well written. This reviewer raises several suggestions which might be better to be addressed before the publication.

Specific comments

1. Introduction might be better to be revised, especially the sentences in Page2, Line 7-15 had not well connected each other. Partly because the three sentences were started from almost same words, such as 1) Recent research trends…… 2) Current research interest is ……., and 3) However recently…  and that makes the reader confused.

2. About the introduction (Page2, Line 19-26), it would be better to emphasize more the rationale why the authors need to review the effect of rosavin in BONE diseases for the society.

3. About the result section (Page3 to 6), basically the authors just summarized two references, #24 and #40. This reviewer understands these two should be key references, however it seems to be too detailed. Especially, the descriptions about #24 are more than a page. It would be better to be revised. For example, it would be better to add more references if possible, and/or omit too detail description (experimental detail or something).  

4. The text of figure2 cannot be seen. Please change the color of box or text.

5. Table 1, 2 would be better to be revised. It is hard to see, partly because too many texts. Also, it would be better to use same font size in the table. 

Round 2

Reviewer 2 Report

Comments and Suggestions for Authors

The authors have included all the required comments.